# Prediction of Type and Recurrence of Atrial Fibrillation after Catheter Ablation via Left Atrial Electroanatomical Voltage Mapping Registration and Multilayer Perceptron Classification: A Retrospective Study

**DOI:** 10.3390/s22114058

**Published:** 2022-05-27

**Authors:** Qiyuan An, Rafe McBeth, Houliang Zhou, Bryan Lawlor, Dan Nguyen, Steve Jiang, Mark S. Link, Yingying Zhu

**Affiliations:** 1Computer Science and Engineering, University of Texas at Arlington, Arlington, TX 76019, USA; houliang.zhou@mavs.uta.edu (H.Z.); yingying.zhu@uta.edu (Y.Z.); 2Medical Artificial Intelligence and Automation Laboratory, Department of Radiation Oncology, University of Texas Southwestern Medical Center, Dallas, TX 75390, USA; rafe.mcbeth@gmail.com (R.M.); bryan.lawlor@utsouthwestern.edu (B.L.); dan.nguyen@utsouthwestern.edu (D.N.); steve.jiang@utsouthwestern.edu (S.J.); mark.link@utsouthwestern.edu (M.S.L.); 3Cardiac Electrophysiology, University of Texas Southwestern Medical Center, Dallas, TX 75390, USA

**Keywords:** registration, atrial fibrillation, electroanatomical voltage mapping

## Abstract

Atrial fibrillation (AF) is a common cardiac arrhythmia and affects one to two percent of the population. In this work, we leverage the three-dimensional atrial endocardial unipolar/bipolar voltage map to predict the AF type and recurrence of AF in 1 year. This problem is challenging for two reasons: (1) the unipolar/bipolar voltages are collected at different locations on the endocardium and the shapes of the endocardium vary widely in different patients, and thus the unipolar/bipolar voltage maps need aligning to the same coordinate; (2) the collected dataset size is very limited. To address these issues, we exploit a pretrained 3D point cloud registration approach and finetune it on left atrial voltage maps to learn the geometric feature and align all voltage maps into the same coordinate. After alignment, we feed the unipolar/bipolar voltages from the registered points into a multilayer perceptron (MLP) classifier to predict whether patients have paroxysmal or persistent AF, and the risk of recurrence of AF in 1 year for patients in sinus rhythm. The experiment shows our method classifies the type and recurrence of AF effectively.

## 1. Introduction

Atrial fibrillation (AF) is the most common sustained cardiac arrhythmia globally [1] and is associated with an increased risk of cardiovascular morbidity and mortality [2]. Restoration of sinus rhythm by catheter ablation has evolved to play an important role in the management of symptomatic AF patients and has been shown to improve cardiovascular outcomes, including mortality, in certain AF patient populations [3]. However, there are still significant recurrence rates of AF after catheter ablation.

The ability to predict recurrent AF after catheter ablation based on predictor models utilizing clinical parameters derived from classical statistical methods is very limited [4]. Atrial remodeling and fibrosis play a central role in the pathogenesis of atrial fibrillation and can be readily evaluated during electroanatomic mapping (EAM) which is performed routinely as part of AF ablation. In EAM, low atrial endocardial bipolar voltage is a commonly used surrogate marker for the presence of atrial fibrosis [5]. The initiation and maintenance of atrial fibrillation (AF) require triggers and an atrial substrate. Low voltage has many useful associations with clinical outcomes, and comorbidities and has links to trigger sites for AF [6]. However, data obtained from EAM are multidimensional and not suitable for standard statistical methods. Neural network methods can analyze higher dimensional data and do not rely on a prior assumption about the underlying relationships between variables. Thus, it would be suitable for analyzing the EAM data to predict recurrent AF after ablation.

The purpose of this study is to retrospectively deploy and evaluate the deep learning methods including PointNet [7], PointNetLK [8] and multilayer perceptron (MLP) to predict recurrent atrial fibrillation after ablation based on an electroanatomical voltage map of patients in sinus rhythm, who underwent AF ablation at the Dallas VA Medical Center (DVAMC). Our experimental results demonstrated the possibility of using PointNetLK to register the electroanatomical voltage maps to a common template. After the registration, we achieved stable prediction of the recurrent AF after ablation on a small dataset.

## 2. Related Work

This section briefly introduces the registration technique and the MLP network for classification.

### 2.1. Point Cloud Registration

Point cloud registration is a fundamental problem in 3D computer vision and photogrammetry. Given several sets of points in different coordinates, registration aims to align all sets of points into a common coordinate [9]. Ref. [10] proposes the iterative closest point (ICP) for registration by iteratively estimating point correspondence and performing least-square optimization. The authors of [7] propose the PointNet for raw point clouds classification and segmentation and provide some insights into dealing with raw point clouds. The PointNetLK [8] combines the PointNet with the classic Lucas & Kanade (LK) algorithm into a trainable recurrent neural network that can generalize across different shapes for point cloud registration.

### 2.2. Multilayer Perceptron

A Multilayer Perceptron (MLP) is a fully connected type of neural network, and consists of at least three layers of nodes, i.e., an input layer, a hidden layer, and an output layer [11]. Trainable weights connect adjacent layers, while there are no connections/weights between nodes within the same layer. For MLPs with more than one layer, we often use some nonlinear activation functions between hidden layers, such as sigmoid, and Relu [12], to increase the model capacity. Training MLPs often employs a supervised learning technique of backpropagation [13]. In our proposed approach, we adopt a two-layer MLP as our classifier.

## 3. Data Collection and Preprocessing

In this study, we utilize AF ablation records of the DVAMC to identify patients aged older than 18 years who underwent AF ablation between 1 January 2015 and 6 January 2019. The DVAMC gave IRB approval for this retrospective study. The Dallas VA electronic medical records (CPRS) of these patients are reviewed to identify inclusion and exclusion criteria, resulting in 12 patients in total. After candidate participants are identified, two datasets from patients are abstracted. First, the EAM mapping data obtained during catheter ablation stored on PACS or local drive are exported as deidentified data in a common separated value file. For this series, all EAMs were mapped with CARTO, using a Pentaray catheter inserted endocardially. Next, the medical information were collected and abstracted from their electronic medical records.

In this work, we used a left atrial electroanatomical voltage map and abstracted clinical information from the cohort to develop algorithms to predict 1-year recurrent AF after AF ablation. For this study, we analyzed only the patients in sinus rhythm at the time of the voltage mapping. Each geometric voltage map is a point cloud with voltages. It keeps the information about the positions and unipolar and bipolar values of 3D points. Each map is related to labels of AF type and recurrence of AF in one year. The AF type includes paroxysmal and persistent sorts. For the 1-year recurrence of the AF label, the criteria are 1-year recurrence of clinically significant AF after a 3-month blanking period, and AF is detected by intermittent ECG (12 lead, commercial devices, e.g., AliveCor Kardia, Apple Watch), continuous ECG (telemetry, Holter, event monitor/Ziopatch, ILR, PPM/ICD) with symptoms, or change in management (rate-control, AAD, DCCV, anticoagulation, EPS, rpt AF ablation, AVJ ablation).

In each map, we use the min–max normalization to normalize the x, y, and z positions and unipolar and bipolar values of point cloud data. We remove maps whose number of points is smaller than 100 and keep all maps recorded in the same ablation, resulting in 8 examples. The number of points in all maps ranges from around 200 to over 4000, and the mean is about 1300. We resample the number of points for each example to 406 without replacement since the lowest number of points is 406 in all eight examples.

Some statistics concerning the data are: all eight patients are male; the age distribution is 1 in 50 s, 3 in 60 s, and 4 in 70 s; the BMI distribution is: two are below 25, two are between 25 and 30, and four are greater than 30.

## 4. Voltage Map Registration and Classification

The overall process of our method is shown in Figure 1, consisting of registration and classification 2 steps.

### 4.1. Registration

Since different patients have different left atrial shapes and thus different electroanatomical maps, we utilize point cloud registration to align the electroanatomical maps among different patients. For voltage map registration, we pretrain a PointNetLK model on the ModelNet40 dataset [14]. Then, we finetune the pretrained model on our electroanatomical voltage maps. We choose leave-one-out cross validation (LOOCV) for finetuning due to the limited number of examples in our dataset. So for each iteration in one epoch, we choose one voltage map as the template $P_{T}$, and the rest are sources $P_{S}$. Our objective seeks to find the rigid-body transform G∈SE(3) [15] that best aligns source $P_{S}$ to template $P_{T}$. Following [8], both source $P_{S}$ and template $P_{T}$ are passed through a shared MLP to compute the global feature vectors $\phi (P_{S})$ and $\phi (P_{T})$. The Jacobian *J* is computed once using $\phi (P_{T})$. The optimal twist parameters are found by Equation (Equation 1), used to incrementally update the pose of $P_{S}$.
(1)$$\begin{array}{c} \xi =J^{+}\left[\phi \left(P_{S}\right)-\phi \left(P_{T}\right)\right] \end{array}$$
where $J^{+}$ is a Moore–Penrose inverse of *J*. Then the global feature vector $\phi (P_{S})$ is recomputed.

In summary, the PointNetLK iterates a looping computation of the optimal twist parameters via Equation (Equation 1), and updates the source voltage map $P_{S}$ as
(2)$$\begin{array}{cc} P_{S} & \leftarrow \Delta G\cdot P_{S}\\ \Delta G & =exp\left(\sum_{i}\xi_{i}T_{i}\right). \end{array}$$

The final estimate $G_{est}$ is the composition of all incremental estimates computed during the iteration:(3)$$\begin{array}{c} G_{est}=\Delta G_{n}\cdot \;...\;\cdot \Delta G_{1}\cdot \Delta G_{0}. \end{array}$$

The training loss function minimizes the difference between the estimated transform $G_{est}$ and the ground truth transform $G_{gt}$
(4)$$\begin{array}{c} L=\left|\right|{\left(G_{est}\right)}^{-1}\cdot G_{gt}-I_{4}{\left|\right|}_{F}, \end{array}$$
following a straightforward method from the representation of $G_{est},G_{gt}\in SE\left(3\right)$. Note that $G_{gt}$ is an optional term. In pretraining, $G_{gt}$ is a randomly generated transform matrix while not needed in finetuning.

### 4.2. Classification

Before feeding voltages into a classifier, we still need to reorder unipolar/bipolar voltages into the same coordinate system before feeding them into the classifier since the sequence/order of voltages has changed with associated 3D points after registration. Therefore, we simply choose to reorder both template and source voltages by their associated points’ L1 distance to the original point as in Equation (Equation 5):(5)$$\begin{array}{c} d_{i}=\left|\right|\left(x_{i},y_{i},z_{i}\right)-\left(0,0,0\right){\left|\right|}_{1}. \end{array}$$

In our task, since we have two types of inputs, unipolar and bipolar, and two prediction outputs, AF type and 1-year recurrence AF (1Y re AF), we then have four independent combinations (unipolar to AF type, bipolar to AF type, unipolar to 1Y re AF, and bipolar to 1Y re AF). In addition, we also propose to use ensemble inputs, i.e., combining unipolar and bipolar voltages, to predict AF type and 1Y re AF. We employ an MLP with two fully connected layers, one dropout layer, and the Relu activation function as the classifier. The loss function is cross-entropy loss with log and softmax function as in Equation (Equation 6)
(6)$$\begin{array}{c} L_{CE}\left(y,t\right)=-\sum_{j}t_{j}\log\left(y_{j}\right)+\left(1-t_{j}\right)\log\left(1-y_{j}\right), \end{array}$$
where $t_{j}$ is the target and $y_{j}$ is the prediction.

In the ensemble method with two types of inputs, i.e., unipolar and bipolar, we merge their representations at the logits layer by averaging the probabilities from two classifiers (unipolar and bipolar), respectively.

## 5. Experiment

### 5.1. Registration

In the PointNet pretraining, the batch size is 16, the epoch is 200, and the training optimizer is Adam [16]. The dimension of feature for each point is 1024. After training in PointNet, we transfer the model into PointNetLK. In the PointNetLK model, the hyperparameters are the same as PointNet. We set the max iteration on LK in each sample to 20 and the step size for approximate Jacobian to 0.01. In the PointNetLK model, we randomly generate a twist vector ξ with six dimensions, in which there are three for rotation and three for translation. This twist vector, serving as the ground truth transformation, will be used to transform the original point cloud data into the source of the registration and the original data will be the template in training to test the performance. We use the random rotation angel [0, 45] degrees and translation [0, 0.8] during training of the PointNetLK model.

The process of registration in the training data is shown in Figure 2. The red point cloud data represents the source point cloud. The blue one is the template. In Figure 2, we choose one training sample to show the registration process evolving from the 1st to the 16th iteration.

### 5.2. Classification

In the classification, as explained in Section 4.2, we train six classifiers for combinations between three inputs (unipolar, bipolar, and ensemble inputs) and two outputs (AF type and 1Y re AF). First, we select one sample as the registration template. Then, we register other samples through the PointnetLK to align other samples to the template. After the registration, we reorder the voltage values of all samples uniformly so that these voltage values can be fed into the classifier. In our implementation, we reorder the voltages according to their 3D points’ L1 distance to the original point. In this way, the order of voltage values will imply the same physical meaning for all samples. After the reordering, we feed the voltage values into an MLP with two fully connected layers to perform classification. Since different samples have a different number of points, we sample them to have the same number of points (406). Our two-layer MLP classifier has a hidden dimension of 300, and the dropout rate is 0.4. The evaluation metrics are accuracy, F1-score, ROC-AUC, and PR-AUC. We perform the leave-one-out cross validation (LOOCV) in all experiments, which uses seven examples for training and one for testing and loop through all examples in the dataset.

Since we have three kinds of inputs (unipolar, bipolar, and ensemble) and two labels (AF type and 1 year recurrence of AF), there are six single evaluation scenarios: unipolar vs. AF type (Table 1), bipolar vs. AF type (Table 2), ensemble vs. AF type (Table 3), unipolar vs. 1Y re AF (Table 4), bipolar vs. 1Y re AF (Table 5), and ensemble vs 1Y re AF (Table 6).

For single input type, we can observe that both kinds of inputs (unipolar (Table 4) and bipolar (Table 5)) can perform very well on two labels. However, the AF type label is more difficult than the 1Y re AF. Under the AF type label from Table 1 and Table 2, we can see that the bipolar input wins slightly against the unipolar input by 3:2. Then, there are three cases in which they are even.

We also compare two ensemble input cases, i.e., we perform weighted average probabilities of two inputs (unipolar and bipolar) for classification. The results are summarized in Table 3 and Table 6. From the ensemble method results, we can observe that the ensemble method in two cases can tie with or outperform the best of single input methods in most cases (mostly the 1Y re AF cases). This scenario indicates that ensemble methods can take advantage of the information from both inputs to perform better in the classification. Only one case underperforms the single input method, i.e., template 0 under the AF type label. This special case is the average of two single input methods and yields that template 0 may cause some disagreement between two types of inputs (unipolar and bipolar).

## 6. Discussion

### 6.1. Unipolar vs. Bipolar

We also compare the performance of two types of input data (unipolar vs. bipolar) to see which one has better classification results. For the 1Y re AF case, we can observe from Table 4 and Table 5 that both inputs can perform very well on the classification. However, for the AF type label, we can see that the bipolar inputs (with three winning cases) perform slightly better than the unipolar inputs (with two winning cases), and there are three even cases.

### 6.2. Discussion of Different Templates

From the classification results discussed in the previous section, we can observe that different templates also play an important role in classification performance, for example, template 2 vs. template 5 in Figure 3. From the given examples of templates 2 and 5, we can see that template 2 can generate better registration than template 5. Therefore, it can also yield better downstream classification results.

### 6.3. Limitation

This work mainly studies utilizing 3D point-cloud registration and MLP to classify voltage mappings for AF type and 1 year recurrence of AF. One obvious limitation is the limited size of our dataset, with only eight examples, and thus the results may have a large bias. In addition, the lack of patients’ demographic information may lead to inferior results in prediction. In our proposed method, we sort the 3D points by their L1 distance to the original point. This design is very heuristic and may not be valid in electroanatomical mapping.

## 7. Conclusions

In this paper, we study the most common cardiac arrhythmia, atrial fibrillation (AF), through classifying patients’ left atrial electroanatomical voltages. To resolve the voltage misalignment problem among different patients, we adopt the 3D point cloud registration method to first align all points (voltage mapping) to the same template. Then we reorder the points of each patient according to the same distance metric. After the registration and reordering, we perform classification on two kinds of input data (unipolar and bipolar). Our experiment results demonstrate that the performances of bipolar values are slightly better than the unipolar values for the classification task, and the ensemble of two inputs can slightly help to improve the classification results in most cases. We hope this work can shed some light on solving similar problems that need point cloud registration.

## Figures and Tables

**Figure 1 sensors-22-04058-f001:**
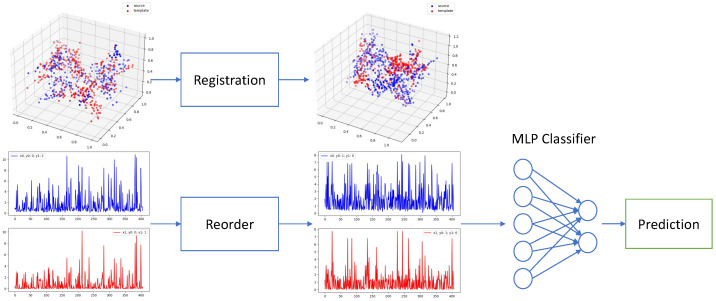
Our proposed registration and classification framework. We first align all EAMs to the same template point cloud via 3D point cloud registration. Then we reorder uni/bipolar voltages of corresponding points by their L1 distance to the original point. We feed the reordered voltages into an MLP for classification in the end.

**Figure 2 sensors-22-04058-f002:**
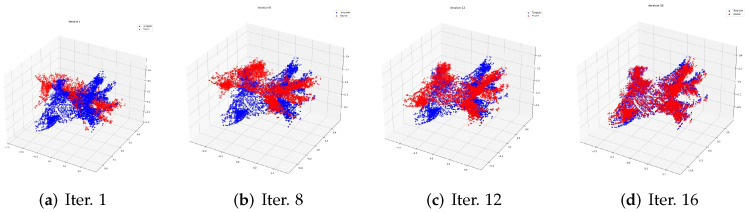
The process of registration in one training data item with patient Id 52997 from the 1st to the 16th iteration. The red point data represent the sources of the process, while the blue represent the template.

**Figure 3 sensors-22-04058-f003:**
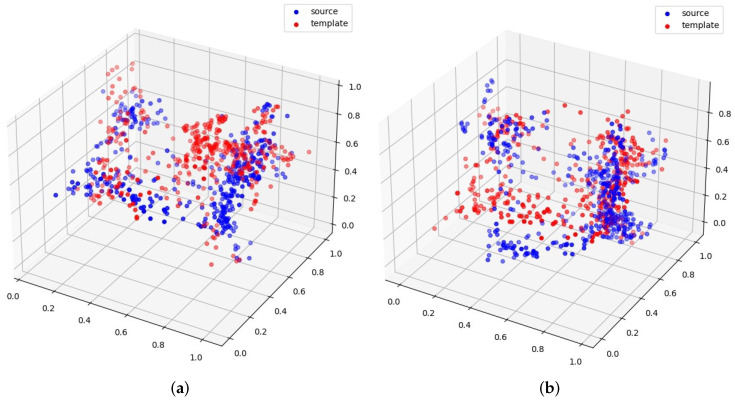
(**a**) Template 2; (**b**) Template 5.

**Table 1 sensors-22-04058-t001:** Classification in eight test data of unipolar voltage on AF type through LOOCV.

Template	Accuracy	F1	ROC-AUC	PR-AUC
0	0.375	0.444	0.375	0.575
1	0.375	0.545	0.375	0.652
2	0.625	0.769	0.500	0.688
3	0.75	0.500	0.667	0.792
4	0.75	0.500	0.667	0.792
5	0.375	0.444	0.375	0.575
6	0.375	0.444	0.375	0.575
7	0.5	0.333	0.467	0.458

**Table 2 sensors-22-04058-t002:** Classification in eight test data of bipolar voltage on AF type through LOOCV.

Template	Accuracy	F1	ROC-AUC	PR-AUC
0	0.625	0.4	0.625	0.8125
1	0.625	0.727	0.625	0.786
2	0.625	0.769	0.500	0.688
3	0.5	0.333	0.467	0.458
4	0.5	0.667	0.4	0.188
5	0.375	0.444	0.375	0.575
6	0.375	0.444	0.375	0.575
7	0.75	0.5	0.667	0.792

**Table 3 sensors-22-04058-t003:** Classification in eight test data of ensemble voltages on AF type through LOOCV.

Template	Accuracy	F1	ROC-AUC	PR-AUC
0	0.5	0.667	0.5	0.75
1	0.625	0.727	0.625	0.786
2	0.625	0.769	0.5	0.688
3	0.875	0.8	0.833	0.896
4	0.75	0.5	0.667	0.792
5	0.375	0.545	0.375	0.652
6	0.5	0.333	0.5	0.562
7	0.75	0.5	0.667	0.792

**Table 4 sensors-22-04058-t004:** Classification in eight test data of unipolar voltage on 1Y re AF through LOOCV.

Template	Accuracy	F1	ROC-AUC	PR-AUC
0	0.75	0.857	0.5	0.875
1	0.75	0.857	0.5	0.875
2	0.875	0.933	0.500	0.938
3	0.75	0.857	0.5	0.875
4	0.75	0.857	0.5	0.875
5	0.625	0.769	0.417	0.836
6	0.625	0.769	0.417	0.836
7	0.875	0.933	0.5	0.938

**Table 5 sensors-22-04058-t005:** Classification in eight test data of bipolar voltage on 1Y re AF through LOOCV.

Template	Accuracy	F1	ROC-AUC	PR-AUC
0	0.625	0.769	0.417	0.836
1	0.75	0.857	0.5	0.875
2	0.875	0.933	0.5	0.938
3	0.5	0.667	0.333	0.792
4	0.75	0.857	0.5	0.875
5	0.625	0.769	0.417	0.836
6	0.5	0.667	0.333	0.792
7	0.875	0.933	0.5	0.938

**Table 6 sensors-22-04058-t006:** Classification in eight test data of ensemble voltages on 1Y re AF through LOOCV.

Template	Accuracy	F1	ROC-AUC	PR-AUC
0	0.75	0.857	0.5	0.875
1	0.75	0.857	0.5	0.875
2	0.875	0.933	0.5	0.938
3	0.75	0.857	0.5	0.875
4	0.75	0.857	0.5	0.875
5	0.75	0.857	0.5	0.875
6	0.75	0.857	0.5	0.875
7	0.875	0.933	0.5	0.938

## Data Availability

Data and code are available at: https://github.com/anqiyuan/left-atrial accessed on 6 April 2022.

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
