# Peer review of "Prediction of Type and Recurrence of Atrial Fibrillation after Catheter Ablation via Left Atrial Electroanatomical Voltage Mapping Registration and Multilayer Perceptron Classification: A Retrospective Study"

_sensors, 2022, doi:10.3390/s22114058_

Round 1
Reviewer 1 Report
I would like to congratulate the authors on the interesting idea of EAVM based Afib classification. I have read with great interest your research paper and in general it is informative and quite easy to read paper. However, there are several serious flaws that definitely should be corrected. The main problem is that you used clinical title of the article completely misguiding your merely thechnical article utilizing neural networks in EAM scenario. There is not enough patients/studies to draw any conclusion regarding classification of Afib type or its recurrence. I suggest you completely re-write the article and set different (non-clinical) goals of the retrospective analysis.
Title:
The title of the manuscript is inappropriate while:
- It says nothing about classification by neuronal networks (MLP), which are the main method in your study
- Moreover it should be noted the retrospective nature of the study
- Where it radiofrequency ablations? Catheter ablations?
Abstract:
- EAM of how many patients were included in the study? You commented on datasize limitations in abstract without mentioning the number of patients/studies. This is actually crucial.
- It should be explained wat is a multilayer perceptron, at least that a neuronal network was used for classifying instead of pinpointing limitations of the study.
- Limitations of the study should be mentioned at the end of the article before conclusion
Introduction:
In the abstract you mentioned that you put both uni- and bipolar in MLP. Why? As you correctly stated in this section, only bipolar data are used as a surrogate for advanced atrial fibrosis.
Materials and Methods:
- Where are the patient data? You used EAM maps from several studies (nowhere in the article mentioned) and moreover you completed omitted not only the number of patients but also their characteristics. Classification of atrial fibrillation could have made sense but with the patient characteristics not without.
- The number of mapping studies involved is important especially due to neural networks ability to learn to discriminate different features of the maps (do i correctly read that you used just 10 maps to feed the neural network), that would be merely a case series.
- Which EAM system was used (CARTO, NavX, Rhythmia)?
- What was the original method of 3D points annotation? Where the points automatically or manually annotated? Because all this features significantly influence the quality of the maps and then all the results
Results:
- Where do you get the information about Afib recurrence? ECG holter or some implantable/external looper was utilized? This is important to assure the validity of the type of atrial fibrillation you are dealing with
Reviewer 2 Report
The paper is devoted to the voltage mapping classification in order to predict the recurrence of atrial fibrillation after ablation. The task is very actual. The authors proposed special preprocessing procedure for EAM data and NN-based classifier. Experimental investigation is done using retrospective data.
The paper consists of short introduction and experimental part. Theoretical description of the idea is poor (which is often allowed for communication). Some portions of text are unclear. It is recommended to add an algorithm or scheme. The part describing classification results is better, but there are some questions too. The references are relevant.
This paper will surely be interesting for researchers in the fields of electroanatomic mapping, NN and AF prediction. But the lack of sufficient details may make practical application difficult. It is recommended to describe registration process more clearly and indicate the number of original maps used in training and testing.
The subject of the article can be ascribed to the subjects of the journal Sensors, specifically the signal processing, data fusion and deep learning in sensor systems.
Some remarks to the authors:
- “…develop algorithms to predict 1-year recurrent AF after AF ablation” (line 51). No formal description of the proposed algorithm
- “For this study we analyzed only the patients in sinus rhythm at the time of the voltage map” (line 52) “mapping”?
- “Each geometric voltage map are point cloud data with voltage” (line 53) ”is”?
- “For each training data, we use data augmentation technique to randomly sample 1024 points from the orginal point clouds” (line 64). “Augmentation technique” requires description in more details.
- Because of imbalanced labels in binary classification we duplicate random data from the minority class during training” (line 65). More details are needed. How many maps were used for training?
- “… to train efficiently for the point cloud registration” (line 69). More details are needed.
- “In the process of registration, the source will be registered to the templale” (line 70) – a misprint in the last word. More details are needed.
- “regietration” (line 71) – misprint
- “trandform” (line 72) – misprint
- “In training, it loops the step of regietration to minimize the loss between estimated trandform and ground truth transform until it hit the max iteration” (line 71). The description of the process is poor.
- “we align all of point cloud data into the same space” (line 73). It is not enough to understand clearly.
- “We choose the top 10 most complete maps as the different templates in registration”. (line 77) How do the authors estimate completeness?
- “We choose the top 10 most complete maps as the different templates in registration for training to calculate the classification result in the test data and detect the sensitivity of the analysis pipeline” (line 77). Training and test data in one sentence make it unclear whether 10 maps were chosen for training or test data.
- Figure 1 and Figure 2 are not very informative.
- “It averages the score of probability in 10 models for each classes as the final classification score” (line 83) – misprint “class”. The formula may help understand the idea.
- “This twist vector will be used to transform the original point cloud data to be the source of the registration and the original data will be the template in training to test the performance” (line 93). Some explanations are needed.
- “we reorder the voltages according to their 3D coordinates’ L1 distance to the original point” (line 110). The scheme may help understand it correctly.
- “Since different samples have different number of points, we sample them to have the same number of points (406)” (line 114). In the previous part “We resample the number of points for each training example to 1024”. Finally 1024 or 406?
- “AF type or 1 year recurrence of AF as the target for the classifier” (line 106). Does it mean that both classifiers have two output states: paroxysmal or persistent (for AF type) and 1 year recurrence of AF or not (1 year recurrence of AF)? It is better to declare it.
Round 2
Reviewer 1 Report
Dear Authors,
you really did a great in improving the merit of the article. Now it is mor technical an in my opinion suited for publication in sensors.
Author Response
Thanks.
Reviewer 2 Report
The paper is devoted to the voltage mapping classification in order to predict the recurrence of atrial fibrillation after ablation. The task is very actual. The authors proposed special preprocessing procedure for EAM data and NN-based classifier. Experimental investigation is done using retrospective data.
The paper consists of short introduction and experimental part. Theoretical description is provided to explain the main steps. The references are relevant.
This paper will surely be interesting for researchers in the fields of electroanatomic mapping, NN and AF prediction. The idea and results are quite clear but it must be kept in mind that the results were obtained on a very small number of original maps used in training and testing.
The subject of the article can be ascribed to the subjects of the journal Sensors, specifically the signal processing, data fusion and deep learning in sensor systems.
Some remarks to the authors:
- Figure 1 could be more informative if some very short explanations of the registration process, reordering and classification result were added. It also could be shown how the process of registration in the first line is connected with the second line. Also I can't see the difference between the two 3D maps.
- It should be clearly declared that two different classifiers with specific outputs are proposed (AF type and 1 year recurrence of AF).
